

# Members of the methanotrophic genus *Methylomarinum* inhabit inland mud pots

Danielle T. Fradet[1], Patricia L. Tavormina[2] and Victoria J. Orphan[2]

[1] Flintridge Sacred Heart Academy, La Canada Flintridge, CA, United States
[2] Geological and Planetary Sciences, California Institute of Technology, Pasadena, CA, United States

## ABSTRACT

Proteobacteria capable of converting the greenhouse gas methane to biomass, energy, and carbon dioxide represent a small but important sink in global methane inventories. Currently, 23 genera of methane oxidizing (methanotrophic) proteobacteria have been described, although many are represented by only a single validly described species. Here we describe a new methanotrophic isolate that shares phenotypic characteristics and phylogenetic relatedness with the marine methanotroph *Methylomarinum vadi*. However, the new isolate derives from a terrestrial saline mud pot at the northern terminus of the Eastern Pacific Rise (EPR). This new cultivar expands our knowledge of the ecology of *Methylomarinum*, ultimately towards a fuller understanding of the role of this genus in global methane cycling.

## INTRODUCTION

The biological conversion of methane to biomass introduces carbon into the food web chemosynthetically. Members of both *Archaea* and *Bacteria* are capable of oxidizing methane for energy and carbon, and represent a significant biological sink for methane on Earth (*Conrad, 2009*; *Reeburgh, 2007*). Among Bacteria, methane oxidation occurs aerobically (reviewed in *Hanson & Hanson (1996)* and *Trotsenko & Murrell (2008)*) with 23 validly-described genera of aerobic *Alphaproteobacteria* and *Gammaproteobacteria* encompassing the majority of cultivated bacterial methane oxidizers (*Knief, 2015*). Twelve of these (*Methyloferula*, *Methylohalobius*, *Methylomarinovum*, *Methylogaea*, *Methylomagnum*, *Methyloparacoccus*, *Methyloglobulus*, *Methylosoma*, *Methylomarinum*, *Methylosphaera*, and *Methyloprofundus*) are represented by a single validly-described species each, and their ecological impact is not clear.

*Methylomarinum vadi* (*Gammaproteobacteria*, family *Methylococcaceae*) was isolated in 2007 from a marine mat associated with a hydrothermal feature near Taketomi Island, Japan (strain IT4, *Hirayama et al., 2013*; *Hirayama et al., 2007*). Taketomi Island lies at the subduction zone of the Philippines and Eurasian plates. Another strain of *Methylomarinum vadi* was isolated independently in 1998 from coastal mud near

Corresponding author
Patricia L. Tavormina,
pattytav@gps.caltech.edu

Hiroshima, Japan (strain T2-1, *Fuse et al., 1998*; *Hirayama et al., 2013*). Because these strains were isolated from locations over 1,300 km apart, *Methylomarinum* has been postulated to be ubiquitous. However, clear molecular signatures for uncultured *Methylomarinum* have only been reported from a single additional site, the brine:seawater interface of the Red Sea (*Abdallah et al., 2014*). The Red Sea overlies the rift zone defined by the Arabian and African plates. Thus, undefined environmental factors, possibly associated with plate boundary zones, may promote the environmental establishment of *Methylomarinum*.

The Eastern Pacific Rise (EPR) is a rift zone that extends from Antarctica to the Salton Sea basin in southern California (Fig. 1). The EPR is predominantly oceanic, and deep-sea hydrothermal vents and chemosynthetic communities occur along its length (e.g. *Baker & Urabe, 1996*; *Shank et al., 1998*). The northern terminus of the EPR occurs on land, in the Salton Sea basin in southern California, and provides a 'terrestrial analog' of deep sea vents and seeps (*Svensen et al., 2009*). The Salton Sea itself is an inland body of water that formed following the diversion of the Colorado River between 1905 and 1907. Due to the combination of agricultural runoff and evaporative loss of water, the sea steadily increases in salinity, which currently stands at 4.4%. The larger Salton Sea basin has notable hydrothermal features including fumaroles, seeps, and mud volcanoes (gryphons) that emit steam, carbon dioxide, and methane (*Mazzini et al., 2011*). The Davis-Schrimpf seep field near the southeast shoreline of the Salton Sea comprises one such cluster of hydrothermal features (Fig. 1).

Despite its unique geology, setting, and history, microbial ecological studies of the Salton Sea region have not been widely reported. Two studies suggest that microorganisms in Salton Sea sediments affiliate with marine lineages (*Dillon, McMath & Trout, 2009*; *Swan et al., 2010*). Microbial ecological studies on the Davis-Schrimpf hydrothermal seep field have not, to our knowledge, been undertaken. We were interested in asking whether methane-oxidizing organisms are part of the microbial community in this field, and if so, whether they are similar to terrestrial or marine methanotrophs. Here we report the first methanotrophic isolate from this region.

## METHODS

### Sampling and enrichment

Two mud pots in the Davis-Schrimpf seep field (latitude 33.200, longitude −115.579) were sampled on 9 September 2012. Temperature and pH were determined for each pot (Table 1). Approximately 400 ml of mud (~30% solids) from each pot was sampled into sterile 1l polycarbonate bottles. The bottles were capped and transported to the laboratory at ambient temperature.

On 12 September 2012, a 10 ml slurry from each bottle was transferred to 30 ml serum bottles, capped with black butyl stoppers, and sealed with aluminum crimps. Methane was added to a final headspace concentration of 50%, and the bottles were kept at 22 °C.

On 30 January 2016, additional samples were collected from 6 mud pots and associated gryphons (mud volcanos less than 3 m in height) in the Davis-Schrimpf seep field (Sites 1–6, Fig. S1) as described above, except that 50 ml polypropylene tubes were used for

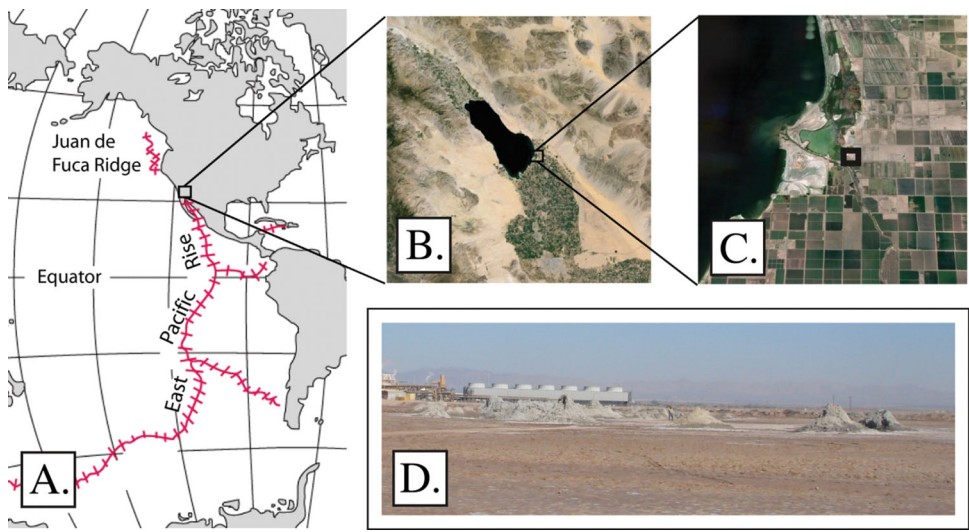

**Figure 1 Study area.** The Davis-Schrimpf seep field occurs at the northern terminus of the Eastern Pacific Rise, and approximately 3 km east of the southeast corner of the Salton Sea. (A) Location and extent of the Eastern Pacific Rise. Image credit: US Geological Survey. (B) The Salton Sea, surrounding basin, and agricultural fields. Image credit: Google Earth, Landsat. (C) The Davis-Schrimpf seep field in relation to the Salton Sea. Image credit: Google Earth, Landsat. (D) A view of the seep field from a southwest approach.

collection, and the collected volume was approximately 30–50 ml per sample. Salinity, temperature, and pH were recorded for each sample (Table 1).

## Media

A modified nitrate mineral salts (modified NMS) media was used for cultivation. This medium is similar to nitrate mineral salts (NMS, ATCC 1306), except that trace elements from DSMZ 141 replaced the trace elements from medium ATCC 1306. Vitamins were added (DSMZ 141, 1,000×), and sodium chloride was added to 2%. The medium was sterilized by filtration. To solidify medium for plates, Bacto agar (Becton Dickinson and Company, Franklin Lakes, NJ, USA) was added to 1.5% final concentration and the medium was autoclaved.

## Isolation of methanotrophic species

In June 2015, approximately 20 μl of slurry from each enrichment bottle was streaked onto modified NMS agar plates. Plates were incubated in a cultivation jar (Oxoid anaerobic jar, Thermo Fisher Scientific, Grand Island, NY, USA) with 50% methane and 50% air in the headspace. Incubations proceeded for fourteen days at 22 °C, and individual colonies of distinct morphology were targeted for additional analysis.

## Molecular identification of methane–oxidizing colonies

Colonies were probed for the presence of the *pmoA* gene (which provides a molecular indicator of methane oxidation potential) via polymerase chain reaction (PCR). Single colonies were picked with sterile pipet tips into 50 μl of sterile phosphate–buffered saline (PBS, 10 mM $Na_2HPO_4$, 1.8 mM $KH_2PO_4$, 137 mM NaCl, 2.7 mM KCL, pH 7.4). Two μl
**Table 1 General features of mud pots and gryphons at the Davis-Schrimpf seep field, January 2016.**

| Site[a] | Sample | Temp. (C) | pH[b] | Salinity (%)[c] | Copies *pmoA* gene/ng DNA (SD) | Copies *pmoA* gene/g sample |
|---|---|---|---|---|---|---|
| 1 | Turbid water | 18.7 | 6.4 | 7.2 | 2.7E + 02 (1.8E + 01) | 1.2E + 05 |
| | Sediment | 18.7 | 6.5 | 7 | 2.2E + 03 (8.5E + 01) | 7.9E + 06 |
| 2 | Southeast end | 18.3 | 6.2 | 5 | **1.4E + 04 (7.7E + 02)** | **6.3E + 07** |
| | Northwest end | 18.3 | 6.3 | 5 | **6.6E + 04 (3.9E + 03)** | **1.5E + 08** |
| | Gryphon | 51.7 | ND | 0.8 | 3.6E + 01 (3.3E + 00) | 3.9E + 03 |
| | 9-2012, mid-pot | 35–37 | 6.83 | ND | ND | ND |
| 3 | Water | 17.2 | 6.2 | 16 | 2.6E + 01 (5.4E + 00) | 3.6E + 03 |
| | Sediment | 17.2 | 6.8 | 17.2 | 9.3E + 02 (1.4E + 02) | 3.5E + 06 |
| | Gryphon | 43.3 | ND | 1.2 | 4.3E + 01 (1.1E + 01) | 3.2E + 03 |
| | 9-2012, mid-pot | 35–37 | 7.07 | ND | ND | ND |
| 4 | Water | 16.1 | 6.7 | 14.2 | 2.8E + 02 (6.8E + 01) | 7.7E + 04 |
| | Sediment | 16.1 | 6.7 | 14.2 | 1.5E + 02 (2.1E + 01) | 3.9E + 04 |
| | Gryphon | 46.9 | ND | 0.2 | 1.7E + 01 (5.4E + 00) | 3.6E + 03 |
| 5 | Sediment | 16.7 | 6.45 | 10.2 | 4.3E + 02 (6.5E + 01) | 3.9E + 05 |
| | Gryphon | 62.8 | ND | 1 | 8.4E + 00 (3.0E − 01) | 1.5E + 04 |
| 6 | Water | 16.7 | 6.76 | 12.4 | 2.6E + 03 (5.5E + 01) | 8.4E + 06 |
| | Sediment | 16.7 | 6.5 | 13 | 1.2E + 03 (1.3E + 02) | 2.4E + 06 |

**Notes:**
ND, not determined.
[a] See Fig. S1.
[b] The pH values of gryphons were not determined due to the high viscosity of these samples.
[c] Salinity was measured in the supernatant derived from pelleted sediment, using a hand held refractometer.

of this suspension was used as a template in PCR; final reaction volume was 20 μl. Reactions were performed in 1× Taq reaction buffer, with 0.2 mM each dNTP, 10 μM of forward and reverse primers (*pmoA*189f and mb661r, *Costello & Lidstrom, 1999*), synthesized by Integrated DNA Technologies, Coralville, IA, USA) and 1 unit Taq polymerase (New England Biolabs, Ipswich, MA). Reactions were heated to 95 °C for 10 min to lyse bacterial cells, and then cycled at 95 °C (1 min), 54 °C (1 min), and 72 °C (1 min). After 30 cycles, a final extension step at 72 °C (1 min) was included. Four μl of the product was electrophoresed through a 1% (w/v) agarose gel, stained with SYBR safe dye (Thermo Fisher Scientific, Grand Island, NY, USA), and visualized with ultraviolet transillumination. Bacterial colonies that gave a PCR amplicon of ~500 base pairs were retained for further analysis.

For determination of 16S rRNA gene sequences, PCR was performed as above except that primers Bac8f and Uni1492r (*Lane, 1991*, synthesized by Integrated DNA Technologies, Coralville, IA, USA) replaced *pmoA*189 and mb661r, and a ~1,400 bp product was expected. When using PCR to amplify target genes from DNA (instead of bacterial cells), the initial denaturation step was reduced from 10 to 3 min.

## Clone library construction

To identify the major contaminating species in an early enrichment culture, PCR primers Bac8f and Uni1492r (*Lane, 1991*) targeting the 16S rRNA gene were used to generate a

~1,400 bp PCR product. This product was purified from 1% agarose gel electrophoresis using the Wizard DNA extraction kit (Promega, Madison, WI, USA) following manufacturer's recommendations. The gel-purified product was cloned into the pSMART GCHK vector using the pSMART system (Lucigen, Middleton, WI, USA) and following manufacturer's recommendations. Forty-eight colonies containing the 16S rRNA gene insert were sequenced commercially.

## Sequencing

To prepare amplicons for sequencing, PCR products were diluted with sterile water to 100 μl, and separated from primers on 96-well filter plates under vacuum (EMD Millipore, Billerica, MA, USA). Cleaned amplicons were resuspended in 30 μl of 10 mM Tris pH 7.5, and DNA concentrations were measured by Nanodrop (Thermo Scientific, Wilmington, DE, USA). DNA amplicons were sequenced commercially (Laragen, Culver City, CA, USA).

## Purification of Methylomarinum strain SSMP-1

To generate a pure culture of *Methylomarinum* strain SSMP-1 (Salton Sea Mud Pots), colonies that gave a positive *pmoA* PCR signal were inoculated into 3 ml of sterile liquid modified NMS and incubated under 50% methane headspaces in Balch style tubes. Inocula were grown with gentle rotation at 22 °C until they reached slight but visible turbidity. These cultures were diluted via a tenfold dilution series to an estimated 1,000 cells per ml (4 dilution steps). At this point in the dilution series, 2 fold dilutions were made past extinction (an additional 14 tubes). This set of 18 tubes was incubated with rotation at 22 °C, and the 'most dilute' tube that ultimately grew was subjected to this same dilution-to-extinction procedure, until molecular (amplification and sequencing of the 16S rRNA gene) and microscopic (fluorescent in situ hybridization) inspection failed to detect any contaminants. At this point *Methylomarinum* strain SSMP-1 was considered pure. The isolate was stored at −80 °C as described in *Hoefman et al. (2012)*.

## Growth determinations

The growth rate of *Methylomarinum* strain SSMP-1 was determined by measuring the change in optical density (450 nm) of liquid cultures grown at 10, 22, 28, 37, and 45 °C. Duplicate tubes were grown at each temperature and results were averaged.

The effect of trace metals on the growth of *Methylomarinum* strain SSMP-1 was determined by measuring the change in optical density (450 nm) of liquid cultures amended with 1×, 2×, 4×, 6×, 8×, 10× and 20× concentrations of DSMZ141 trace element solution. Duplicate tubes were grown for each amended media, and results were averaged.

## DNA extraction

To extract DNA from cell cultures, 4–10 ml cultures at an optical density of 0.3 were pelleted in polypropylene tubes in a Beckman Coulter Allegra X-15R tabletop centrifuge at 4,000 RPM for 20 min. The supernatant was discarded and the cell pellets were subjected
to lysozyme-proteinase K treatment followed by a phenol-chloroform extraction protocol and ethanol precipitation (*Wilson, 2001*). The resulting DNA was treated with an additional cesium chloride and ethidium bromide cleanup step (*Saunders & Burke, 1990*), to attain a level of purity suitable for molecular analysis.

To extract DNA from environmental samples, the Power Soil DNA extraction kit was used following manufacturer's specifications (MO BIO Laboratories, Carlsbad, CA, USA). Approximately 0.2 g of sediment was extracted per sample. A FastPrep bead beater (Bio101, Thermo Fisher Scientific, Waltham, MA, USA) was used to disrupt cells during this procedure.

## Microscopy: FISH and motility

To visualize cells via fluorescence in situ hybridization (FISH), 0.5 ml of exponential-phase culture was fixed with 2% paraformaldehyde (PFA, 2 h, 22 °C). Standard protocols were used to perform FISH (e.g. *Manz et al., 1992*). In brief, fixed cells were transferred onto 20 × 64 mm glass slides and hybridized with 16S rRNA-targeted fluorescently-labeled oligonucleotide probes specific for all bacteria (Fluos-labelled EUB 338 I-III; *Amann et al., 1990*; *Daims et al., 1999*) and *Gammaproteobacterial* methanotrophs (Cy3-labelled Mγ705; *Eller, Stubner & Frenzel, 2001*). Primers were synthesized by Integrated DNA Technologies (Coralville, IA, USA). Following hybridization, cellular DNA was stained with diamidino-2-phenylindole (DAPI). Cells were imaged on an Olympus BX51 microscope under oil immersion at 1,000× magnification.

To observe motility in *Methylomarinum* strain SSMP-1, log phase cells were removed from their stoppered vessel with an 18-gauge needle attached to a 3 ml syringe. A drop (~25 μl) of this culture was deposited gently onto a glass slide and covered with a 20 × 60 mm coverslip. This wet-mounted preparation was viewed using phase contrast on an Olympus BX51 microscope under oil immersion at 1,000× magnification.

## Quantitative PCR

To enumerate *Methylomarinum* from cultures and environmental samples, qPCR primers specific to *Methylomarinum* were designed (*Methylomarinum_pmoA267*f: ACTRTTTCTGTATTRGCGCTG, and *Methylomarinum pmoA*375r: AAYCARGTTTGAT GGGAATACG). Primers were synthesized by Integrated DNA Technologies, Coralville, IA, USA. At least 1 mismatch in each primer is predicted for all other validly published methanotrophic genera, thus, as a paired set, these primers are not expected to easily amplify from methanotrophs outside of *Methylomarinum*. The primers were empirically validated in this regard against genomic DNA corresponding to *Methylomarinum* strain SSMP-1, *Methylosinus trichosporium* strain OB3b (Genbank accession number AF186586), *Methylocystis parvus* strain OBBP (AF533665), *Methylomicrobium album* strain BG8 (FJ713039), *Methylosarcina lacus* strain LW14 (AY007286), *Methylosarcina fibrata* (AF177325), *Methylomonas* sp. strain LW13 (AF150793), *Methylomonas methanica* strain S1 (U31653), *Methylomicrobium alcaliphilum* strain 2OZ (FO082060), *Methyloprofundus sedimenti* strain WF1 (KF484908), and from the nitrosifiers *Nitrosococcus oceani* (U96611) and *Nitrosomonas europaea* (AF037107). Optimal annealing temperature for this primer pair was 53 °C. The 'no template control'

used PCR grade water in place of DNA template, and generated a Cq of 39 or greater in quantitative PCR. DNA templates from environmental samples were assayed at 1-fold, 4-fold, and 16-fold to verify that the assay fell within the linear range (i.e. no inhibitory substances in the DNA extract). The standard curve was generated against 10 fold dilutions of genomic DNA from *Methylomarinum* strain SSMP-1. Assuming a *pmoA* copy number of 1 and genome size of $4.3 \times 10^6$ bp, which is the size of the *Methylomarinum vadi* strain IT4 genome, the standard curve ranged from 20 to $2 \times 10^7$ copies per microliter. The dynamic range of the assay encompassed these values.

Quantitative PCR was performed using the Power SYBR Universal mastermix (Thermo Fisher Scientific, Grand Island, NY, USA) and a Biorad CFX96 Real time C1000 Touch Thermal Cycler System. Prior to cycling, DNA was denatured for 3 min at 95 °C. Then, forty cycles of qPCR (denaturing at 95 °C, for 15 s and annealing and extending at 57 °C for 1 min) were performed. A dissociative melt curve (0.5° per minute) was included to evaluate amplicon homogeneity. Measurements were made in triplicate and averaged; standard deviations were calculated. The products of qPCR were sequenced commercially (Laragen, Culver City, CA, USA).

## Phylogenetic analysis

For the 16S rRNA gene, sequences from select species within *Methylococcaceae* and relevant environmental sequences were used to infer a tree by maximum-likelihood, using the PhyML package (*Guindon et al., 2010*) and the HKY evolutionary model in the software program ARB (version 5.5, http://www.arb-home.de/; *Ludwig et al., 2004*). The sequences were aligned in the SSURef-119-SILVA-NR database (http://www.arb-silva.de/; *Quast et al., 2013*) and masked using the provided bacterial filter. The tree's reliability was estimated by bootstrapping in software program Geneious version 7.1.7 (http://www.geneious.com/) using PhyML maximum-likelihood, the HKY model and 1,000 replicates. Sequences that formed the outgroup (not shown) included *Methylocapsa acidiphila* strain B2 (AJ278726), *Methylocystis parvus* strain OBBP (Y18945) and *Methylosinus trichosporium* strain OB3b (NR_044947).

To reconstruct a phylogenetic tree for the A subunit of particulate methane monooxygenase, pMMO-A sequences from cultured methanotrophs within *Methylococcaceae*, *Methylothermaceae*, and select environmental sequences were aligned using MUSCLE (http://www.phylogeny.fr; *Dereeper et al., 2008*). The resulting alignment was analyzed via maximum-likelihood, with the AMO-A sequences of *Nitrosomonas cryotolerans* ATCC 49181 (AF314753) and *Nitrosospira briensis* strain C-128 (U76553) as the outgroup (not shown). Sequences generated in this study were deposited to Genbank under accession numbers KU740209–KU740211 and KX060584.

## RESULTS

### Isolation of Methylomarinum strain SSMP-1

We routinely place environmental samples under methane, as one means of building resources for investigating microbial methane oxidation. Two such incubations, from

samples collected at the Davis-Schrimpf seep field (Fig. 1) near the southeast shore of the Salton Sea, were established in September 2012.

In June 2015, a small amount of sediment slurry from each of these incubations was inoculated onto solid modified NMS medium and placed under 50% methane headspace. After 14 days, approximately 22 distinct colony types were identified that ranged in size, coloration, and shape. These colonies were probed for the particulate methane monooxygenase gene (*pmoA*), whose product, pMMO-A, catalyzes aerobic methane oxidation. Of the 22 colonies probed, two generated strong *pmoA* amplicons. Both colonies derived from the "Site 2" mud pot (Fig. S1). The sequences of these two amplicons were identical, and were 89% identical (at the DNA level) to the *pmoA* sequence of the marine methanotroph *Methylomarinum vadi* IT-4.

Both colonies were streaked three times successively on agar in an effort to purify them from co-cultivating bacteria; however, this approach did not yield a pure culture. One of the colonies was inoculated into liquid media and purified via the dilution-to-extinction method twice, until no contaminating species were evident (Fig. 2). At this point, both the *pmoA* and the 16S ribosomal genes were amplified from the culture. The 16S rRNA gene demonstrated 96.7% identity to *Methylomarinum vadi* IT-4, and the pMMO-A protein scores were 97% (identity) and 99% (similarity) when compared to *Methylomarinum vadi* IT-4 (Fig. 3). These results place strain SSMP-1 within *Methylococcaceae*, and within *Methylomarinum*, in *Gammaproteobacteria*.

## Characteristics of Methylomarinum strain SSMP-1

The doubling time of *Methylomarinum* strain SSMP-1 in liquid modified NMS was measured at 10, 22, 28, 37, and 45 °C. The strain did not grow at 10, 37, or 45 °C. At 22 °C, the strain doubled every 15 h, while at 30 °C the strain doubled every 18 h. These results indicate a lower optimal growth temperature for *Methylomarinum* strain SSMP-1 than that reported for *Methylomarinum vadi* IT-4, which has an optimal growth temperature of 37–43 °C (*Hirayama et al., 2013*). As *Methylomarinum* strain SSMP-1 approached saturation, it formed large flocs, in sheets approximately 3–4 mm across. Because hydrothermal features often exhibit elevated metal concentrations, the tolerance of *Methylomarinum* strain SSMP-1 to elevated metals was preliminarily assessed, by amending culture media with up to 20-fold higher trace element solution. Even at the highest end of this amendment series, *Methylomarinum* strain SSMP-1 grew with near normal kinetics (not shown). In contrast, the reference marine organism *Methyloprofundus sedimenti* (*Tavormina et al., 2015*) was inhibited for growth at a 6-fold elevation in trace element concentration.

When grown on solid modified NMS media, *Methylomarinum* strain SSMP-1 formed colonies within two weeks that were approximately ½ mm across. Colonies were smooth and shiny at two weeks but rough and irregular at four weeks. Non-uniform pink pigmentation developed as incubations proceeded past 2 weeks (Fig. S2).

Individual cells of *Methylomarinum* strain SSMP-1 were short, thick rods approximately 1.3–1.5 μm long (Fig. 2), similar to the size reported for *Methylomarinum vadi* IT-4 (*Hirayama et al., 2013*). At a magnification of 1,000×, cells appeared

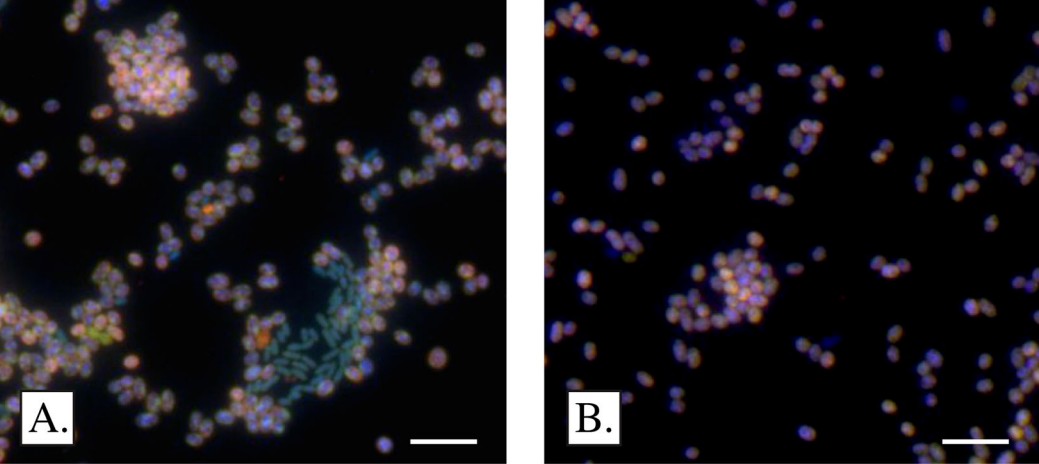

**Figure 2 Microscopic appearance of *Methylomarinum* strain SSMP-1.** (A) Rod shaped bacteria initially co-purified with the isolate, and were typically surrounded by cells of *Methylomarinum* strain SSMP-1. (B) No contaminating bacteria were evident after two successive dilution-to-extinctions in liquid modified NMS. Scale bar: 5 microns.

granular (Fig. 2). Cells were clearly motile (Movie S1), in keeping with reported features of the type species of the genus (*Hirayama et al., 2013*). During the course of purifying *Methylomarinum* strain SSMP-1, cells were routinely imaged with FISH. These images indicated that *Methylomarinum* strain SSMP-1 initially existed in close association with smaller rod-shaped bacteria (Fig. 2A). These rods were typically surrounded by *Methylomarinum* strain SSMP-1, but occasionally occurred independently.

The contaminating rods were not detected in the pure culture (Fig. 2B). A 16S rRNA clone library derived from the enrichment culture revealed that approximately 83% of 16S rRNA genes affiliated with *Methylomarinum*, whereas 16% affiliated with genus *Methylophaga* (96.9% identity to accession KF790925; *Mishamandani, Gutierrez & Aitken, 2014*). This finding suggests that organisms capable of utilizing methanol as a carbon source may associate with *Methylomarinum* in the Davis-Schrimpf seep field environment.

## Natural abundance of Methylomarinum in the Davis-Schrimpf seep field

Because *Methylomarinum* strain SSMP-1 was isolated following a three-year enrichment at room temperature, its significance to the in situ microbial community in the seep field is not clear. To assess the natural abundance of *Methylomarinum* in the seep field, DNA from six mud pots and associated gryphons was extracted and probed directly for the *pmoA* gene using quantitative PCR with primers specific to *Methylomarinum*. These primers did not amplify DNA from ten reference organisms (D. Fradet, 2016, unpublished data). Environmental samples for this analysis were collected in January 2016, after four years of drought in California, and the water level of the mud pots was significantly reduced relative to September 2012, when the original samples were collected (Fig. S3). Additionally, the samples were collected in

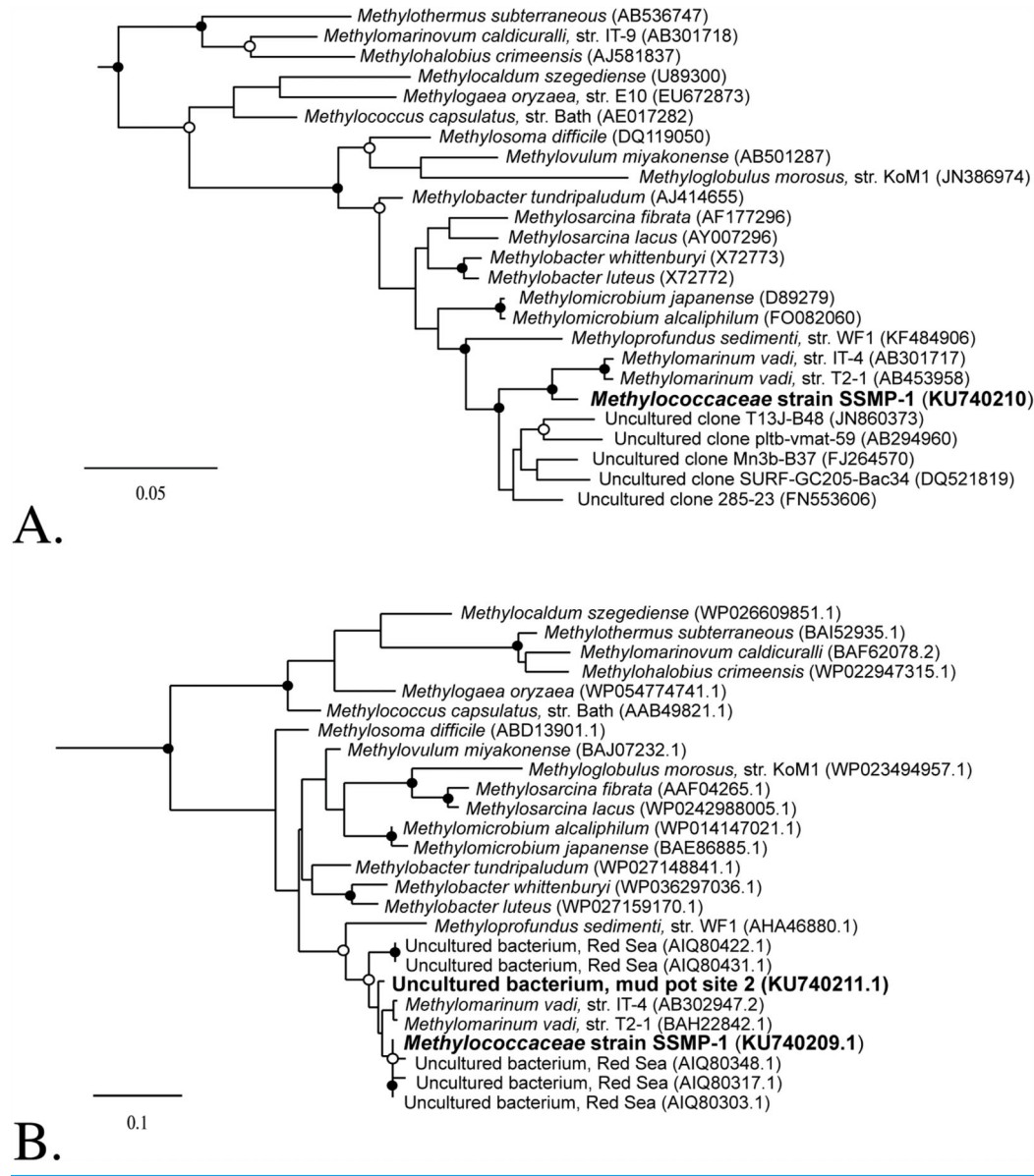

**Figure 3 Phylogenetic relatedness of *Methylococcaceae* strain SSMP-1 to select methanotrophic genera.** Closed circles indicate bootstrap value greater than 80%. Open circles indicate bootstrap values greater than 60%. (A) A 16S rRNA gene phylogeny. Analysis of the 16S rRNA gene supports placement of *Methylococcaceae* strain SSMP-1 near *Methylomarinum vadi* strains IT4 and T2-1. (B) A pMMO-A protein phylogeny. Analysis of protein sequences from select methanotrophic genera also indicates that *Methylococcaceae* strain SSMP-1 affiliates with *Methylomarinum vadi*. Sequences amplified directly from Site 2 mud pot DNA, and sequences from uncultured organisms at the brine:seawater interface in the Red Sea, are shown.

January when ambient temperatures are near their lowest values of the year. Despite these distinctions between the two collection dates in this study, *Methylomarinum pmoA* genes were readily detected in all mud pots, and were most abundant at Site 2, the original source of *Methylomarinum* strain SSMP-1 (Table 1). At this site they were detected at over $10^4$ copies per nanogram of DNA. Sequencing these amplicons

confirmed that they are closely related to *Methylomarinum*, although several base changes exist between in situ amplified *pmoA* sequences and the *Methylomarinum* strain SSMP-1 *pmoA* sequence. A near-complete *pmoA* gene sequence was recovered from Site 2 DNA, to verify affiliation with *Methylomarinum* (Fig. 3).

A general trend existed between increasing salinity of the mud pots and decreased abundance of *Methylomarinum,* as assessed by abundance of the *pmoA* gene (Table 1). Also, a clear distinction in abundance existed between gryphons and mud pots. In gryphons, abundance of *Methylomarinum* did not exceed ~50 copies per nanogram of DNA; this equates to three orders of magnitude less than the abundance of *Methylomarinum* in the Site 2 pot. The salinity of gryphons was between 0 and 1.2 in all cases, lower than the salinity in mud pots where values were between 5.0 and 17.2 (Table 1). The reported tolerated salinity range of *Methylomarinum vadi* is 1.0–8.0, with an optimum of 2.5–3.0 (*Hirayama et al., 2013*), which fits reasonably well with the numbers reported here, from individual pots and gryphons within the seep field.

## DISCUSSION

The fumaroles, mud pots, and gryphons in the Salton Sea basin are emission sources of one-carbon compounds, including methane and carbon dioxide. The Davis-Schrimpf seep field provides a terrestrial analog of submarine hydrothermal features in the Guaymas Basin, within the Gulf of California (*Svensen et al., 2009*). Yet, methane (and carbon dioxide) cycling in the Davis-Schrimpf field is essentially uncharacterized. The present work demonstrates, firstly, that methane-oxidizing bacteria reside in abundance in the Davis-Schrimpf seep field; secondly, that the ecological range of genus *Methylomarinum* includes inland mud pots; and thirdly, that the Davis-Schrimpf seep field shares not only morphological and geochemical elements with deep-sea hydrothermal features, but also microbial ecological elements.

Hydrothermal features along seismic zones source $CO_2$ and $CH_4$ to the environment, and the biological constraints that limit the escape of these gases to the atmosphere are not fully characterized. In deep marine systems, methane emissions are naturally remediated by sediment–hosted Archaea and Bacteria. Methane that escapes sediments can be oxidized within the water column, which hosts a distinct methane–oxidizing assemblage below 200 m water depth (*Tavormina, Ussler & Orphan, 2008*; *Tavormina et al., 2013*). Methane-oxidizing microbial assemblages in shallow ocean waters have yet to be clearly or definitively identified. Within the Davis-Schrimpf seep field, where methane is sourced from shallow saline water, methane that is not remediated in situ is expected to directly enter the atmosphere. Thus, methane-oxidizing assemblages in the mud pots within this field are of particular interest, for their role in limiting the escape of methane to the atmosphere.

Both *Methylomarinum vadi* strain IT-4 and *Methylomarinum* strain SSMP-1 were isolated from hydrothermal features, along seismic zones. *Methylomarinum vadi* strain T2-1 was isolated from mud at several meters water depth, near Hiroshima, Japan.

Molecular signatures (*pmoA*) of related uncultured organisms (Fig. 3 this work; *Abdallah et al., 2014*) have been reported from the seawater:brine interface in the Atlantis II Deep brine pool in the Red Sea, at a water depth of approximately 2,000 m. Red Sea brine pools are characterized by hypersalinity, low oxygen, acidity, and elevated concentrations of metals (*Bougouffa et al., 2013*). Molecular signatures (16S rRNA) with slightly lower relatedness to the *Methylomarinum* 16S rRNA gene have been detected from hydrothermal vent sites, sediment incubations amended with manganese, and from hypersaline sediments from the Gulf of Mexico (Fig. 3; *Beal, House & Orphan, 2009*; *Hirayama et al., 2007*; *Li et al., 2013*; *Lloyd, Lapham & Teske, 2006*; *Schauer et al., 2011*). Taken together, the available evidence may suggest that genus *Methylomarinum* preferentially establishes in environments characterized by elevated salinity or temperature, reduced pH, and/or high metal content. Initial queries here into the roles of temperature, salinity, and metals suggest that genus *Methylomarinum* may be particularly tolerant of elevated metal concentrations. Indeed, analysis of genomic potential among available methanotrophic genomes reveals that the number of genes conferring resistance to cadmium, zinc, and cobalt ranges from a low of 17 in *Methylohalobius* to a high of 34 in *Methylococcus*. Importantly, *Methylomarinum* also encodes 34 genes relevant to cadmium, zinc, and cobalt tolerance (D. Fradet, 2016, unpublished data). This result places *Methylomarinum* at the uppermost end of the range. Direct quantitation of *Methylomarinum* from habitats ranging in salinity, pH, and methane and metal concentrations, will likely provide additional insight into the physicochemical or geological factors that promote the establishment of this genus in the environment. Such studies, coupled with controlled laboratory investigations with cultured members of *Methylomarinum*, will ultimately improve our understanding of the role of this genus in global methane cycling.

## ACKNOWLEDGEMENTS

We thank Roland Hatzenpichler and Derek Smith for sampling the Salton Sea mudpots in 2012 and providing critical methodological, graphic, and scientific input. We thank Alyssa Boedigheimer for assistance in sample collection in January 2016. We thank Stephanie Connon for assistance with the 16S rRNA gene phylogeny.

### Funding

Funding for this work was provided by the Gordon and Betty Moore Foundation, in a grant to Victoria J. Orphan (grant no. GBMF3780). This research was additionally supported by a grant from the NASA Astrobiology Institute (Award # NNA13AA92A). This is NAI-Life Underground Publication Number 083. The funders had no role in study design, data collection and analysis, decision to publish, or preparation of the manuscript.

## Grant Disclosures

The following grant information was disclosed by the authors:
Gordon and Betty Moore Foundation, in a grant to Victoria J. Orphan: GBMF3780.
NASA Astrobiology Institute, in a grant to Victoria J. Orphan: NNA13AA92A.

## Competing Interests

The authors declare that they have no competing interests.

## Author Contributions

- Danielle T. Fradet conceived and designed the experiments, performed the experiments, analyzed the data, wrote the paper, reviewed drafts of the paper.
- Patricia L. Tavormina conceived and designed the experiments, analyzed the data, wrote the paper, prepared figures and/or tables, reviewed drafts of the paper.
- Victoria J. Orphan conceived and designed the experiments, wrote the paper, reviewed drafts of the paper, intellectual Support, encouragement, and funding.

## DNA Deposition

The following information was supplied regarding the deposition of DNA sequences:
Genbank Accession numbers KU740209–KU740211, and KX060584.

## Data Deposition

The raw data has been supplied as Supplemental Dataset Files.

## Supplemental Information

Supplemental information for this article can be found online at http://dx.doi.org/10.7717/peerj.2116#supplemental-information.

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
