# Peer review of "Members of the methanotrophic genus Methylomarinum inhabit inland mud pots"

_PeerJ, doi:10.7717/peerj.2116_

## Round 0.1 · original submission · Minor Revisions

· Academic Editor

Minor Revisions

Both of the reviewers would clearly favour publication of the paper in Peer J, but they have asked for specific revisions (and pointed towards some discrepancies and anomalies), which I would ask you to address in full.

Also, I think it would strengthen the paper considerably if you could include any additional information or data that would help to define more precisely the geochemical conditions that would favour growth of Methylomarinium.

Reviewer 1 ·

Basic reporting

This is a well-written and -presented manuscript that describes a study in which methanotrophic bacteria were isolated from a terrestrial mud pot field site. Based on 16S rRNA gene and pmoA gene sequence data the isolate labelled SSMP-1was shown to be sufficiently closely related to Methylomarinum vadi, the only species in the Methylomarinum genus, to be considered a representative of this genus, which previously have only been detected in/isolated from marine sites.

The manuscript's introduction clearly sets the context of the work using relevant references.

Figures, tables and the movie are all of good quality and well described, with only minor comments (see attached pdf, Table 1 annotation).

Raw data did not seem to be supplied, but that would probably related to only the sequencing traces, but accession numbers for sequences are provided.

I am not sure whether the essential components of the qPCR standard information (as on http://miqe.gene-quantification.info/) were all supplied (eg calibration parameters).

Experimental design

The research question is well defined, and the experimental approaches of the study appear sound and have been well described. As far as I know (and as discussed by the authors), this is the first description of methanotrophs in this particular geological feature (Davis-Schrimpf hydrothermal seep field), and due to the fact that these directly vent CH4 to the atmosphere (without a surrounding water column), it is indeed of interest to identify whether methanotrophs are present that may mitigate some of the flux to the atmosphere.

Validity of the findings

The main findings are that
- Methylomarinum related bacteria are present in the mud pots
- there are methanotrophs in this geological feature

The authors conclude that the prevailing geochemical conditions of high salt, reduced pH and perhaps high metal content may be factors that are drivers for the colonisation of this habitat by Methylomarinum. The data presented all suggest that this may indeed be so. There is no claim about the dominance of these bacteria or the importance of these bacteria in situ (as well as potential role of other methanotrophs that may be present). In that sense, the results are a novel finding, that provides new original insights into the potential role of Methylomarinum spp in methane cycling in this type of environment, as well as setting the scene for follow on studies that need to assess methanotroph diversity and activity in this environment more fully.

Annotated reviews are not available for download in order to protect the identity of reviewers who chose to remain anonymous.

Reviewer 2 ·

Basic reporting

The main claim of this paper is demonstration, through pure culture isolation and culture-independent identification, of members of the genus Methylomarinum in inland mudpots. While it is not necessarily surprising as the site in question obviously originated from the neighboring sea, I think the discovery is of value. Little is known about communities and activities in these environmental niches.

Experimental design

The experimental design is straightforward and follows the state of the art in the field.

Validity of the findings

The evidence presented seems to be solid.

Additional comments

General comments:
The title of the paper says Methylomarinum, and indeed all the evidence points toward the new species described and the sequences detected via PCR to belong to Methylomarinum. However, in Results and Discussion (and Figures), the authors switch to Methylococaceae. I do not see a reason for such discrepancies. I think the organisms should be referred to Methylomarinum and not otherwise.

A very interesting fact is reported, of association of Methylomarinum with another bacterium, as shown in Fig. 2a. Can you tell us who this bacterium is? You probably could easily identify it, or do you already have this information? This could bring an interesting dimension toward understanding community function in these interesting environments?

Specific comments:
Line 27. While it may seem a little strange, methane is officially an organic compound, so it makes no sense to claim that methane is converted into organics.

Line 230, line 238. Why do you get 87% and 99% for the same sequence? You need to explain these discrepancies.

Line 240. And within Methylomarinum.

Line 288. Salinity may be one factor, but the temperature may be another factor?

Line 327. Of course, such assumptions should not be made based on one single strain, especially since you could not determine such of the parameters for comparisons, such as the pH value.

---

## Round 0.2 · accepted · Accept

· Academic Editor

Accept

Many thanks for revising the paper in accordance with the reviewers' comments. Reviewer 2 has suggested some further minor revisions, which seem sensible to me, and I would be most grateful if you would consider these and make appropriate changes to the final version of the paper prior to publication.

I am very pleased to see this paper accepted for publication in PeerJ, and hope that you will consider making future submissions to our journal. Best wishes - Andy Weightman

Reviewer 1 ·

Basic reporting

In the interest of brevity, there is no need to reiterate my positive comments about the manuscript. The authors have responded to the specific comments and suggestions that I made in the intitial review. Indeed the additional insight into metal resistance is an interesting addition.

Experimental design

as before

Validity of the findings

as before

Additional comments

It would be great to actually include a table with results on metal work in this ms, but I would understand that the authors may carry out more extensive work on that aspect. A note should accompany the discussion of metal resistance genes to indicate that this is based on unpublished work and that the data are not included.

Line: 41: should be strain not species (especially if using same species name the former does not make sense

Reviewer 2 ·

Basic reporting

No Comments

Experimental design

No Comments

Validity of the findings

No Comments

Additional comments

I find the changes made to the original manuscript satisfactory. I have no further comments on the revised manuscript.